# If the Yedoma thaws, will we notice? Quantifying detection limits of top-down methane monitoring infrastructures

Martijn M. T. A. Pallandt<sup>1,2,3</sup>, Abhishek Chatterjee<sup>2,4</sup>, Lesley E. Ott<sup>2</sup>, Julia Marshall<sup>5</sup>, Mathias Göckede<sup>1</sup>

- <sup>1</sup>Max Planck Institute for Biogeochemistry, Jena, 07745, GER
- 5 <sup>2</sup>NASA Global Modeling and Assimilation Office (GMAO), Goddard Space Flight Center, Greenbelt, MD 20771, USA
  - <sup>3</sup>Department of Physical Geography and Bolin Centre for Climate Research, Stockholm University, 114 19 Stockholm, SWE
  - <sup>4</sup>NASA Jet Propulsion Laboratory, California Institute of Technology, Pasadena, CA 91011, USA
  - <sup>5</sup>Deutsches Zentrum für Luft- und Raumfahrt, Institut für Physik der Atmosphäre, Oberpfaffenhofen, 82234, Germany
- 10 Correspondence to: Martijn Pallandt (Martijn.Pallandt@natgeo.su.se)

#### **Abstract**

Large quantities of carbon are stored in Yedoma permafrost. When temperatures rise, its high ice content is a catalyst for rapid degradation, which in turn may cause the release of large quantities of carbon. 40% to 70% of the radiative forcing from this release is expected to be in the form of CH<sub>4</sub>. In this observing system simulation experiment, we examined the capabilities of three atmospheric GHG monitoring platforms i.e. tall towers, and the TROPOMI and MERLIN satellite instruments, to detect changes in CH<sub>4</sub> release from increased Yedoma thaw. A set of environments are simulated with the GEOS-5 model: one representing a 'natural' emission case as the reference, a second featuring enhanced CH<sub>4</sub> release from Yedoma soils. From within these modelled environments, synthetic measurements are generated following best in situ practices and realistic error characterizations.

For the satellites we find the lowest detection limits when aggregating measurements over a 112 day period, at Yedoma fluxes of 144% to 367% of current conditions. These factors are up to 1.2 times higher when taking transport modelling uncertainties into account. The tall tower network shows a wide range of detection lower limits, the lowest at only 107% of current fluxes, but has considerably higher lower detection limits when factoring in transport modelling errors. Overall, the individual systems appear to lack the ability to detect and attribute small changes in Yedoma CH<sub>4</sub> fluxes, and would either need to be used in combination or require a considerable time to detect changes under higher emission scenarios.

# 1 Introduction

The Northern high latitudes are seeing rapid changes in environmental conditions as a result of climate change (Serreze and Barry, 2011; IPCC, 2014; Meredith et al., 2019). These changes can have far-reaching consequences since permafrost soils contain large stocks of carbon, almost twice that of the atmosphere (Yu, 2012; Schuur et al., 2013; Hugelius et al., 2014; Strauss et al., 2017; Nichols and Peteet, 2019; Mishra et al., 2021). This carbon may be released to the atmosphere when permafrost thaws (Hugelius et al., 2020; Schuur et al., 2015, 2008; Serreze and Barry, 2011). The form in which this carbon is released (e.g. as carbon dioxide (CO<sub>2</sub>) or methane (CH<sub>4</sub>)) has a large influence on its climate impact (Schneider von Deimling et al., 2015; Walter Anthony et al., 2018), with 40-70% percent of the radiative forcing from permafrost thaw projected to originate from CH<sub>4</sub> emissions. To understand how the Arctic will be affected and to properly capture any changes, continuous monitoring is essential; however, the monitoring capacity for pan-Arctic methane fluxes is still limited (O'Connor et al., 2010; Pallandt et al., 2021; Peltola et al., 2019; Pirk et al., 2016; Wille et al., 2008; Wittig et al., 2023; Xu et al., 2016), and likely not sufficient to detect abrupt changes in methane emissions at an adequate resolution and precision to inform adaptation measures designed by policymakers.

There are many methods to directly monitor the methane exchange processes between the surface and the atmosphere. Bottom-up methods, which include flux chambers and eddy covariance stations, measure locally with footprints ranging from <1 to several 1000s of m² (Pirk et al., 2016; Schimel, 1995; Virkkala et al., 2018; Zona et al., 2016). These measurements can be upscaled to a larger domain to obtain regional-scale methane budgets (Davidson et al., 2017; Ingle et al., 2023; Nelson et al., 2024; See et al., 2024). There are also transient methods such as drone and airborne campaigns (Fix et al., 2023; Miller and Dinardo, 2012; Scheller et al., 2022; Shaw et al., 2021; Sweeney et al., 2022), capable of capturing the spatial variability of methane flux signals in the atmosphere in episodic snapshots. Top-down methods make use of observations over large regions based on greenhouse gas sensors mounted on tall towers or satellites. To relate changes in measured atmospheric concentrations to fluxes between the biosphere and atmosphere, atmospheric inverse modelling is a commonly-used technique (Houweling et al., 2017; Michalak et al., 2004; Miller et al., 2014; Peters et al., 2010; Rödenbeck et al., 2003; Thompson et al., 2017). In atmospheric inverse modelling, emissions are estimated by minimising a cost function that compares observed

atmospheric mixing ratios with simulated values based on surface-atmosphere fluxes and transport models, including estimates of related uncertainty fields. Details on these methods can vary (Brasseur and Jacob, 2017), while the result is usually some form of local to regional estimate of fluxes constrained by observed concentrations.

# 1.1 Tall towers

However, all top-down methods rely upon measurements of the atmospheric mixing ratios, either via in situ sampling or remote sensing. In this study we are using tall tower measurements to represent in situ measurements in general. Tall towers are typically equipped with in-situ greenhouse gas sensors that allow them to directly sample GHG mixing ratios. Many of these towers take samples from different heights, which corresponds to probing air with increasingly remote origins. Some towers are tall enough to breach the atmospheric boundary layer (at least at night), and take samples from the free troposphere, without a direct link to nearby surface fluxes (Bakwin et al., 1995; Winderlich et al., 2010). Continuous measurements often utilise cavity ring-down spectrometers to constantly sample the air from tower inlets, though they are more limited in the species they can detect (Andrews et al., 2014; Ball and Jones, 2003; Winderlich et al., 2010). As an alternative to direct in-situ GHG measurements, air samples can be collected in flasks at regular intervals, often (bi-)weekly, and stored for later analysis in a laboratory. This method has a lower temporal resolution compared to in-situ analysers, but allows for a large range of compounds and isotopes to be detected (Andrews et al., 2014; Keeling et al., 1976; Levin et al., 2020). Tall towers can have footprints covering several 1000s of km², therefore a single site can capture the influence of surface signals on a regional scale. A network of multiple towers can be used in inversions to link atmospheric concentrations to ground processes.

# 1.2 Satellites

While a tall tower takes measurements at a fixed point within the lower atmosphere, satellites sample the total atmospheric column, with measurements distributed across the globe. Satellite retrievals make use of molecular absorption lines at specific wavelengths to deduce the mixing ratio of a target gas, such as methane. While instruments measuring emitted radiation in the thermal infrared are mostly sensitive to methane in the upper troposphere and lower stratosphere, sensors measuring in the shortwave infrared have sensitivity to the full atmospheric column, making these sensors better able to capture the spatiotemporal variability near the surface, which is important for flux inversions. In the past there have been several missions with such instruments: these include the SCanning Imaging Absorption spectroMeter for Atmospheric CHartographY (SCIAMACHY) on ESA's Envisat (Bovensmann et al., 1999; Buchwitz et al., 2006; Burrows et al., 1995; Dils et al., 2006; Frankenberg et al., 2006), the Japanese GOSAT mission (Butz et al., 2011; Yokota et al., 2009), and the TROPOspheric Monitoring Instrument (TROPOMI) on ESA's Sentinel-5 Precursor mission (Hu et al., 2018; Lorente et al., 2021; Veefkind et al., 2012).

## 1.2.1 TROPOMI

In this study we will take a closer look at TROPOMI's detection capabilities as a state-of-the-art (Lindqvist et al., 2024) passive Short Wave InfraRed (SWIR) sensor with the best spatial coverage. TROPOMI measures in the ultraviolet and visible (270-500 nm), near-infrared (675-775 nm) and shortwave infrared (2305-2385 nm) spectral bands. It is therefore able to detect a host of compounds (e.g. nitrogen dioxide, ozone, formaldehyde, sulphur dioxide, methane and carbon monoxide). It has a spatial resolution as high as 7 km × 5.5 km at nadir. With a swath width of 2600 km and 14 sun-synchronous orbits a day, it produces a large number of soundings, especially around the poles where soundings overlap with those from previous orbits. A radiative transfer model is used to estimate the spectrum that would be expected at the top of the atmosphere based on a prior estimate of the atmospheric state, simulating the instrument sampling. Then pre-defined fit parameters (i.e. the state vector) influencing the atmospheric profile are adjusted, taking into account the uncertainty on the prior guess, to best match the measured spectrum, in a method known as optimal estimation. In the Weighting Function Modified Differential Optical Absorption Spectroscopy retrieval (WFMD), the fit parameters include scaling factors for the prior columns of CH<sub>4</sub>, CO, and H<sub>2</sub>O, a shift parameter for the temperature profile, a scaling factor for the pressure profile, and parameters for a third-order polynomial to fit the sun-normalized radiance, which is a factor of Rayleigh scattering, aerosol optical depth and surface albedo. One of the largest sources of errors in this measurement technique is related to uncertainties in the light path due to scattering from aerosol particles and thin cirrus clouds. Such scattering may both shorten and lengthen the true path length of the light, leading to systematic errors that are difficult to correct, leading to under- and overestimation of XCH<sub>4</sub>. Further, as a passive sensor, it cannot sample without sunlight, making its use in the wintertime Arctic limited.

#### **1.2.2 MERLIN**










One way to overcome this innate limitation of passive remote sensing is the use of an active sensor, which comes equipped with its own radiation source, making it independent of sunlight. The French Centre national d'études spatiales (CNES) and the German Aerospace Center (DLR) are developing such a sensor for the Methane Remote Sensing Lidar Mission (MERLIN), which is the sole instrument on the MERLIN mission. with an expected launch date of 2028. The instrument is an integratedpath differential absorption nadir-viewing Lidar (IPAD). Two spectrally narrow laser pulses at frequencies close to 1.64 microns are emitted in the nadir direction in close succession. One pulse has a frequency located in the wing of a pressurebroadened CH4 absorption line, ensuring absorption close to the Earth's surface. The other pulse is located "offline", with negligible CH4 absorption, serving as a reference. The two pulses follow in close succession (250 microseconds apart) to measure nearly identical air masses. In contrast to passive sensors, the light path is well known from the timing of the return pulse, making the measurement much less sensitive to aerosol- and cloud-related errors (Ehret et al., 2017; Pierangelo, C. et al., 2016; Stephan et al., 2011). Thus, the systematic errors are expected to be considerably less than for passive instruments (Bousquet et al., 2018). However, the random error of a single measurement is much larger than for a passive sensor, and multiple single-shot pairs (with a ~150 m diameter, separated by ~650 m) will be averaged in on-ground processing to reduce this, with a nominal averaging length of 50 km, or 142 shot pairs. These random errors depend mainly on the signal intensity measured by the instrument, and are thus negatively affected by low albedo and aerosol scattering. Of note for this high latitude study is that snow has a low albedo in the 1.64 microns range. While TROPOMI offers many more measurements per orbit, MERLIN provides global sampling over the entire year. The difference is especially striking at high latitudes, where TROPOMI is essentially blind through much of the Arctic winter. This study aims to quantify the effect of this difference on the detection of flux signals in the atmosphere.

# 1.3 Inversion modelling

The large regional to global scales on which these systems operate might mean that local effects or processes with small flux magnitudes may remain undetected. While tall towers have large footprints, these are too sparse to cover all regional processes in the domain of interest. Processes relatively close to a tower may be hidden from it due to prevailing winds from different sectors (Pöhlker et al., 2019), and even a signal that falls within the footprint may not be detected as its influence to the final concentration decreases over distance (Vermeulen et al., 2011). Furthermore, inversion models typically report significant transport modelling errors, especially at high northern latitudes (Baker et al., 2006), further complicating the precise spatial attribution of an atmospheric signal. Satellites typically have global coverage, but they still require transport modelling, and calibration against ground-based reference datasets, such as those provided by the Total Carbon Column Observing Network (TCCON) (Wunch et al., 2011), to relate the column-integrated concentrations to ground processes (Bergamaschi et al., 2009; Parker et al., 2011; Toon et al., 2009). Moreover, atmospheric conditions, such as the presence of clouds or aerosols, which are typically detrimental to satellite soundings, need to be considered (Alexe et al., 2015; Bergamaschi et al., 2009; Houweling et al., 2014). As a consequence, each of the available observation platforms features uncertainties that may compromise its ability to monitor minor changes in surface-atmosphere exchange processes.

## 1.4 Outline

To better understand the detection limits of the tall tower network and satellites, we conducted an observing system simulation experiment (OSSE) (Arnold and Dey, 1986; Errico et al., 2013; Zeng et al., 2020) where we tested a scenario of increased CH<sub>4</sub> release from so-called Yedoma soils in the Arctic, and how it would be detected by the three observation platforms introduced above. OSSEs are typically used to test large networks like these where local experiments would not yield meaningful results or for systems that are not yet operational (such as MERLIN). In an OSSE, an environment is modelled that mimics a natural system, and synthetic measurements with realistic errors are generated. Such synthetic measurements can then be compared between a baseline run reflecting in situ conditions and a scenario run where specific conditions are created. Here we use the Goddard Earth Observing System (GEOS) model for these simulations, a framework which is well suited to simulate earth observing missions. The remainder of the manuscript is laid out as follows. Section 2 outlines the Methods, including the model setup for conducting the OSSEs and the simulated sampling strategy for our three observing systems – tall towers, TROPOMI and MERLIN. Section 3 provides the results, specifically focusing on the capability of these sensors to capture various attributes related to detection of methane emissions from Yedoma thaw. We continue with a discussion in Section 4, including caveats associated with our study and summarize the results and findings in Section 5.

#### 145 **2 Methods**




#### **2.1 GEOS**

The Goddard Earth Observing System (GEOS) Earth System Model (Molod et al., 2015; Rienecker et al., 2011) is a versatile coupled ocean-land-atmosphere modelling framework consisting of several components that allow it to address a wide range of questions related to Earth Science investigations. With land, ocean and atmospheric components and the ability to assimilate data for all three of these, it sees a wide range of uses. Particularly relevant for this study is its ability to model the carbon cycle (Ott et al., 2015; Sweeney et al., 2022; Weir et al., 2021), and its use for generating model simulations for OSSEs for satellite signal detection studies (Errico et al., 2013; McCarty et al., 2021).

In this model setup, we simulated CH<sub>4</sub> fields at  $0.5^{\circ}$  horizontal resolution and with 72 vertical layers (up to  $\sim 0.1$  hPa) at a three-hourly temporal resolution. The extend of the model is global though our analysis is limited to north of  $50^{\circ}$  North . CH<sub>4</sub> flux input fields consist of five datasets: (1) agricultural emissions, (2) anthropogenic biofuel emissions, and (3) industrial and fossil fuel emissions, all taken from the Emissions Database for Global Atmospheric Research (EDGAR v4.3.2) (Janssens-Maenhout et al., 2019); (4) biomass burning emissions from the Quick Fire Emissions Dataset (QFED) (Koster et al., 2015); and finally, (5) wetland emissions from the process-based ecosystem Lund–Potsdam–Jena model, WSL version (LPJ-wsl) (Poulter et al., 2011; Zhang et al., 2016). The setup is in line with Sweeney et al. (2022).

160 As a scenario to test the detection limits we focus on expected increased CH<sub>4</sub> release from thawing Yedoma (Schneider von Deimling et al., 2015; Strauss et al., 2017). Yedoma deposits are ice- and carbon-rich permafrost soils which are widespread in Siberia and Alaska (covering more than 10<sup>7</sup> km<sup>2</sup>). These soils are highly vulnerable to disturbance and degradation and are also prone to abrupt thaw processes such as e.g. thermokarst. Strauss et al. (2017) predict that 5-40 TgC from deep sources will be released in the form of CH<sub>4</sub> per year over the coming century. We generate a nature run across an entire year (in this 165 study, we picked the year 2010 for our baseline year), and a high-emission scenario run for the same time period. In this highemission scenario, wetland CH<sub>4</sub> fluxes in grids flagged as containing Yedoma (Fig. 1) are amplified above the baseline from March until the end of the year with all other fluxes unaltered; we call this the flux enhancement (Fe) factor. In this study, the maximum flux enhancement factor applied was 111 (Fig. 2), which was derived by comparing the magnitude of methane emissions from the labile carbon pool at the end of the century (Schneider von Deimling et al., 2015; Strauss et al., 2017) 170 relative to the magnitude of methane fluxes for the year 2010 based on fluxes prescribed in the LPJ-wsl model. However, the main focus was placed on sensitivity experiments whereby the flux enhancement factor was gradually decreased to a minimum value of 1.06 (see Section 2.5). The spatial extent of the Yedoma fields has been adapted from Strauss et al. (2016) (Fig. 1) to match the GEOS 0.5° resolution.

Figure 1: Spatial extent of areas within the Arctic study domain dominated by Yedoma soils (cyan shading), including site locations of the tall tower network (coloured circles). Yedoma areas were adapted from Strauss et al. (2016) to match the GEOS grid resolution of  $0.5^{\circ}$ . 63 tall towers are shown, colour-coded by distance to the closest Yedoma area in km. Land-sea boundary vectors were taken from natural earth.

Figure 2: Tall-tower network wide average methane concentrations for the year 2010 showing the baseline concentrations from the nature run (Blue) and concentrations resulting from an 1.06 and 111-times flux enhancement (Fe) from natural sources in Yedoma

areas (Peach and Yellow respectively). For signal detection the Fe was decreased by a factor of 1.1 in 80 steps down to a lowest value of 1.06.

#### 2.2 Tall-tower network

For this model setup, we identified 63 tall towers in the boreal and Arctic domain located between 42.6 to 82.5 degrees North (Fig. 1, Table S1), for which we designed a realistic synthetic sampling protocol. There is a large variation in elevation above sea level within this network, with the lowest point at Kjolnes (KJN) in Northern Norway 5 metres above sea level and Summit (SUM) at the apex of Greenland's ice sheet at 3215 metres. Concerning the instrument height above ground level, Russia features both the lowest and highest mounting positions with Teriberka (TER) at 2 meters above ground level and ZOTTO (ZOT) with a height of 301 metres. ZOTTO is the only tower that samples in the second atmospheric layer of the GEOS model. Of the 63 towers, 17 are listed to have flask samples with a predominant sampling scheme of one flask per week. Continuous sampling with in-situ gas analyzers was confirmed at 35 of the 63 sites. The exact method of sampling is unknown for the remaining 11 sites. Even in cases where data were collected continuously throughout the day, for this study we restricted the database to samples taken during the day when the boundary layer is expected to be well mixed; however, during the Arctic winter, when very stable stratification dominates, this may still not always be the case.

From the GEOS nature and enhanced flux run, each grid and level that contained a tall tower was sampled 3 times a day from the top inlet height during the 10:00 to 18:00 local time window, with 3 hours between each sample for a total of 1095 samples per site per year. Since GEOS outputs have a 3-hour temporal resolution the time offset to UTC (in hours) from where a tower is located determines if the first sample is taken at 10:00, 11;00 or 12:00. This is in line with typical practice to sample in the afternoon when a well-mixed boundary layer has been formed. The timestep at which GEOS model output was written out was 3-hourly, even though the model internal timestep is much higher.

Two error schemes were applied to the synthetic data:

- Tall Tower ideal scenario (TT<sub>i</sub>) only takes into account the 2 ppb precision as set by the WMO for atmospheric CH<sub>4</sub> sample analysis. This precision error term has a gaussian distribution and is scaled in such a way that 95% of this distribution falls within this -2ppb to 2ppb range (μ 0 ppb, σ 1.02 ppb). This error represents a theoretical detection limit of an atmospheric signal, including the ability to detect a change, but excluding an attribution of the source of the signal.
- In the Tall Tower full error scheme ( $TT_f$ ), a transport modelling error is added to the before-mentioned precision. The transport modelling error represented additional uncertainty caused by a model to link measured concentrations to surface fluxes often distant in space and time from the point of measuring. This transport modelling error is scaled to have a mean absolute error of 30 ppb ( $\mu$  0 ppb,  $\sigma$  44.5 ppb) to match Bergamaschi et al. (2022). This reflects the network's ability to detect a change and attribute it to the region of origin. In this scenario the total error equals ( $\mu$  0 ppb,  $\sigma$  1.02 ppb) + ( $\mu$  0 ppb,  $\sigma$  44.5 ppb).

# 2.3 TROPOMI






For cloud screening of all satellite soundings we used the International Satellite Cloud Climatology Project (ISCCP-H series) dataset (Rossow et al., 2022; Young et al., 2018).

Total-column soundings were generated to match optimal TROPOMI sampling, with one full 227-orbit repeat cycle (~16 days) repeated over the year. To estimate appropriate thresholds for simulating the cloud screening, we looked at the statistics for "good" soundings from two TROPOMI products,: the RemoTeC (v2.0.4) retrieval from the Netherlands Institute for Space Research (hereafter referred to as SRON) (Lorente et al., 2021, 2023) and the Weighting Function Modified Differential Optical Absorption Spectroscopy (2.0) from the University of Bremen (referred to as WFMD) (Schneising et al., 2019, 2023). For the analysis of the WFMD data the standard quality screening was used (xch4\_quality\_flag=0 for good retrievals) and for the SRON v2.0.4 product the screening (qa\_value>0.5) was applied.. We used these to establish general cutoffs and relations between variables and errors. After an initial analysis we solely used the WFMD product since the reported uncertainties in the SRON product appear to be too low, and do not match the scatter when compared with TCCON colocations, unlike the WFMD reported uncertainties. We diagnosed the Solar Zenith Angle (SZA) and established a cutoff at <75°. An assessment of the relation between measurement uncertainty and different factors indicated that albedo dominates. Random errors in ppb, the precision, were modelled by fitting a curve to the reported uncertainties from the WFMD soundings (Fig. S1), showing the strongest relation to SZA and retrieved albedo at 2.3  $\mu$ m (function 1). We binned the data onto our 0.5° x 0.5° model grid for comparison with samples from the model, but counted all soundings (not just one per bin per orbit). A host of filter settings were compared to the measurement coverage from 2020 and 2021 to produce the best fit between the spatial and temporal distribution of their good-quality measurements and our synthetic sampling, resulting in the following filter settings: an ISCCP cloud fraction (cf) < 0.2, solar zenith angle (SZA)  $< 75^{\circ}$ , albedo<sub>2.3 $\mu$ m</sub> x cos(SZA) > 0.01 Land fraction > 0.9 or sea ice fraction >0.995. Pressure-weighted column averaging was applied as averaging kernel to generate these modelled samples (Supplementary Fig. S3).

$$Precision = 10.03 + \frac{0.234}{albedo \cdot cos(SZA) + 0.0034}(1)$$







To apply this relationship to the simulated data, albedo from MODIS band 7 was used (albedo7), which is measured at 2.1  $\mu$ m. For most applications over land, this small difference in spectral albedo should not be significant. However, the MODIS albedo sampled by GEOS is a snow-cleared value available only over land, and does not reflect the snow and ice coverage, where the albedo is set to 0.05. Considering the fraction of the pixel covered with (sea)ice and snow FrI, this results in the following albedo for pixels over land:

$$albedo = (1 - FrI) \cdot albedo7 + FrI \cdot 0.05 (2)$$

Over sea ice (defined as FrI > 0.8 and a land fraction FrL < 0.1), albedo is set to 0.05. Over open water (FrI < 0.8 and FrL < 0.1), only retrievals near the sun glint point are possible, which is negligible at these latitudes (Schneising et al., 2023). To account for the correlation between nearby measurements, the ~3 million soundings were binned by taking the mean of all soundings within 100 km and 1 hour of each other, yielding ~475 thousand samples that were then treated as independent (Figures 3 and 4).

Seven subsets of TROPOMI data were created to investigate the impact of ground conditions on measurement precision (Table 1). The 'Full' subset contains all soundings. 'Sea', 'Ice', and 'Land' contain only soundings from grids with 100% coverage of their corresponding type. The '>0.5' cases contain soundings with at least 50% of the grid covered by their corresponding type, and less than 95% ice or snow (in line with the 0.95 ice cutoff employed in Kiemle et al. (2014)). Note that TROPOMI can only measure over open water when pixels are located close to the sun glint location, which seldom occurs north of 40°.

Table 1 Synthetic satellite retrieval subsets and basic descriptives. FrL indicates the land fraction and FrI the snow and ice fraction. Soundings are the total number of soundings north of 50 degrees latitude. Precision indicates the mean random error per system for the entire domain over one year. The last two rows indicate the MERLIN scenarios including transport modelling errors.

| Subset              | Condition            | TROPOMI   |                 |                 |           |                    | MERLIN                                   |
|---------------------|----------------------|-----------|-----------------|-----------------|-----------|--------------------|------------------------------------------|
|                     |                      | Soundings | Mean XCH4 (ppb) | Precision (ppb) | Soundings | Mean XCH4 (ppb)    | Precision (ppb)                          |
| Full                | All data             | 475932    | 1803            | 18.1            | 588037    | 1803               | 108                                      |
| Sea                 | FrL= 0 & FrI = 0     | 0         | 1               | -               | 60475     | 1809               | 95.4                                     |
| Ice                 | FrI = 1              | 159026    | 1799            | 20.2            | 232279    | 1799               | 155                                      |
| Land                | FrL = 1 & FrI =0     | 9891      | 1807            | 13.1            | 12352     | 1807               | 35.8                                     |
| Sea > 0.5           | FrL <0.5 & FrI <0.95 | 270499    | 1805            | 24.2            | 175645    | 1805               | 74.8                                     |
| Ice > 0.5           | FrI >0.5             | 241590    | 1800            | 20.4            | 373108    | 1801               | 134                                      |
| Land > 0.5          | FrL >0.5 & FrI <0.95 | 229808    | 1807            | 15.2            | 115401    | 1808               | 44.1                                     |
|                     |                      |           |                 |                 | Soundings | Mean XCH4<br>(ppb) | Precision + transport<br>modelling error |
| Full transport Low  | All data             | -         | -               | -               | 588037    | 1803               | 116                                      |
| Full transport High | All data             | -         | -               | -               | 588037    | 1803               | 131                                      |

#### 2.4 MERLIN

Total column soundings were generated to match the planned MERLIN orbits. We filtered out all fully clouded soundings, yielding 589 thousand samples. The random error characterisation is based on the work by Bousquet et al. (2018), and each sample consists of all the along track samples with an averaging length of 50 km (Figures 3 and 4)., with the following precision:

$$Precision = \frac{\sqrt{K_{on}^2 + K_{off}^2}}{\sqrt{142 \cdot (1 - cf)} \cdot 1780} (3)$$

Where

$$K_{on} = \frac{\sqrt{a + b \cdot c \cdot E \cdot D}}{c \cdot D} (4)$$

$$K_{off} = \frac{\sqrt{a + b \cdot c \cdot E}}{c \cdot E} (5)$$

With




280 
$$D = e^{-2 \cdot DAOD_{ref} \frac{P_{surf}}{P_{ref}}}$$
(6)

$$E = e^{-2 \cdot AOD_{SW}} \cdot \frac{\rho}{\rho_{ref}} (7)$$

Here a, b and c are constants which were set to 20, 0.2 and 70 respectively to match Fig. S2b in the supporting information document of Bousquet et al. (2018); cf denotes the cloud fraction, which was taken from the ISCCP data product sampled along the simulated orbit. 142 is the number of shot-pairs that are averaged over the 50-km sampling distance. By multiplying it by (1-cf), the number is reduced by the fraction that would be screened by clouds.  $DAOD_{ref}$  is the Differential Absorption Optical Depth reference value of 0.534 at a CH<sub>4</sub> concentration of 1780 ppb;  $\rho_{ref}$  is the reflectance reference value of 0.1, with  $\rho$  being the reflectance converted from albedo as in Kiemle et al. (2014);  $P_{ref}$  is the standard pressure at sea level at 1013 hPa and  $P_{surf}$  the surface pressure in hPa; finally AOD<sub>sw</sub> is the aerosol optical depth at 1650 nm converted from the AOD at 550 nm (sampled online from the GEOS-5 data assimilation) with the Junge power law (Zhu et al., 2018). The main factors determining the precision are the surface reflectance, cloud fraction, aerosol optical depth and pressure. Just as for TROPOMI pressure-weighted column averaging was applied as averaging kernel to generate these modelled samples (Supplement S3). In addition to the seven subsets we introduced for the TROPOMI sampling, we considered two more scenarios which included transport modelling errors taken from Bousquet et al. (2018). Full transport low reflects the low end of the random error increase as a result of including a transport modelling error of 8 ppb, and full transport high represents the high end of the transport-modelling-related random error at 23 ppb (Table 1).

Figure 3: In the two top tiles mean monthly XCH<sub>4</sub> simulated retrievals from TROPOMI and MERLIN in March, with the precision in the bottom tiles. These values are based on filtered and the spatio temporal averaged retrievals as described in section 2.3 and 2.4. While TROMPOMI nets less unique measurements is has better (lower) precision than MERLIN.

Figure 4: In the two top tiles mean monthly XCH<sub>4</sub> simulated retrievals from TROPOMI and MERLIN in September, with the precision in the bottom tiles. These values are based on filtered and the spatio temporal averaged retrievals as described in section 2.3 and 2.4. While TROMPOMI nets less unique measurements is has better (lower) precision than MERLIN.

# 310 2.5 Signal detection



We compare the nature run with the Yedoma thaw scenario for each of the seven sampling and error characterizations listed above using an array of two-tailed t-tests to detect any difference with the alternative hypothesis that no detectable differences are present. We opted for two-tailed t-tests since in reality we would not know if at a certain point or time a flux would increase or decrease. The basis for the signal detection experiment is a variable signal strength, where we reduce the 111 (Fe) over 80 steps to a minimum Fe of 1.06. Since the power of a t-test increases with sample size, we test six temporal bin sizes of increasing length (7, 14, 28, 56, 112, 224 days) with step sizes of one day as these move across the year, similar to a moving average. However, because of the large number of tests, we can expect a large number of false positives. Therefore we apply a false discovery rate (FDR) correction (Benjamini and Hochberg, 1995) on the p-values and report the resulting q-values. In this context, we consider each q-value of 0.05 or lower to be a significant detection of differences. In the tall tower network, we test each tower individually and report the number of towers that show a significant difference. We then establish the lower

detection limit cutoff point for each time step. The first step is finding the range of Fe values where both significant and non-significant values are present. The top of this range is the lowest significant Fe where all preceding steps were also significant. The bottom of this range is the highest Fe for which all lower Fe values were not significant. The cutoff point is then the centre of this range, weighted by the number of (non)significant values in this range. For example, if this range had three significant and four non-significant results, the cutoff point would be set at three down from the top.

No systematic biases are included since they would be unaffected by the perturbation and thus not affect the outcome of this test.

#### 3 Results






## 3.1 Optimal detection limits

Evaluating the effect of the bin sizes in both the tall tower network (Fig. 3) and satellite systems (Fig. 6) shows that a longer evaluation period increases the discriminatory power of these systems, though only to a certain extent. In the tall tower network and the MERIN subsets *ice* >0.5 and *sea* >0.5, only minimal improvements and in some cases decreases in performance are found after 112 days. In the case of TROPOMI and the remaining MERLIN subsets, no substantial improvements are found in bin sizes longer than 28-56 days. Increases in bin sizes come at a cost, as periods considerably longer than the 112 days will increasingly sample from outside the peak fluxes in this experiment (Fig. 2). And in the case of the satellites, larger bin sizes can increase the number of samples in unfavourable conditions. The Arctic night greatly reduces the data yield from TROPOMI (Fig. 7), and snow and ice negatively affect the precision of both instruments, especially MERLIN (Table 1). Unless noted otherwise, reported values in the text reflect the 112-day bins.

We find that the tall tower network is capable of detecting the lowest flux differences: this happens at the peak of fluxes in September, with a Fe value of 1.07, but only for a single site (Baranov), when excluding transport modelling errors. Including such errors increases the lower detection bound to a flux enhancement Fe of 4.56. We also observe that there is a large difference in detection limits between the towers: For all 63 towers to detect a change, Fe needs to be at least 1.58 ( $TT_i$ ), or up to 32.9 ( $TT_f$ ) when considering transport modelling errors. For at least five towers to detect a significant difference between natural and enhanced emissions, detection limits are more than doubled compared to that for a single site (compare Figures 5 and 8).

TROPOMI's lowest detection limit is slightly higher at an Fe of 1.40 in the Land > 0.5 compared to the Full subset at 1.53. Even at a short 7-day bin size, TROPOMI can detect significant differences at a Fe of ~2.7 under the Full and Land > 0.5 subsets.

In the *Full* subset, MERLIN's lowest detection limit is at an Fe value of 3.67; however, the Land > 0.5 subset performs better than the *Full* subset at an Fe of 3.21, owing to significantly lower random errors over land (Table 1). The impact of transport modelling errors appears to be relatively small, with the Low scenario in some cases having similar detection limits as the *Full* scenario without transport modelling errors and the High scenario only adding an average 0.76 Fe to the detection limit.

Figure 5: Minimum detection limit ranges for the tall tower network between detection at a single site (lower bound) and at all sites of the network (upper bound) per temporal bin size. The non linear y-axis shows flux enhancement (Fe). Without transport

Figure 6: Minimal detection limits for MERLIN and TROPOMI by temporal bin size. The Y-axis shows the Fe values of the lowest detection limits (note the shorter Y-axis and linear scale). On the X-axis the size of temporal bins is given. Full sets are in magenta, Sea in purple, Ice in cyan and Land in yellow. Distinctions between MERLIN and TROPOMI and the subset thresholds are shown in line style. We see optimal detection limits at a temporal bin size of 112 days. A version with all cases can be found in Supplementary Figure S4.

Figure 7: Number of cloud-screened synthetic satellite soundings, by month,  $5^{\circ}$  latitude bin and platform, left: TROPOMI, right: MERLIN. TROPOMI is not capable of reliable soundings over clear water at these latitudes. Because of the precession of its orbit MERLIN will not sample north of  $85^{\circ}$  N.

# 3.2. Sea Ice and Land subsets

For both satellite-based platforms we investigated the effect of different surface conditions on the retrievals by splitting the dataset by Sea, Ice or Land grids, and grids that predominantly contained one of these classes.

Since TROPOMI has no reliable way of sampling over open sea at these latitudes, the Sea > 0.5 subsets therefore reflect samples taken from predominantly sea grids containing land or ice. The Sea > 0.5 and Ice > 0.5 subsets performed worse than the Full subset at a Fe of 3.43 and 2.51 respectively (Fig. 6). The Land >0.5 performed slightly better at 1.40. However, the Ice and Land subsets performed significantly worse than the Full subset, at a Fe of 5.31, and 3.94 respectively. In the case of Ice this is likely a result of a strong seasonality in the sampling, since it has a similar number of total samples and mean error to the Land > 0.5 subset (Table 1). The difference is that the flux enhancement is low in the winter months when ice and snow dominate (Fig. 1).

Owing to the large sensitivity to ground conditions, we see large differences between the subsets in MERLIN data. The lower detection limit for Sea, Ice, and Land is ~7 Fe higher than for the Full subset. The detection limits of these scenarios are fairly similar since the number of samples and mean errors are proportional, with Land having the smallest sample size and the smallest error and Ice the highest number of samples and the largest error (Table 1). In the >0.5 subsets we see that sample size is no longer the limiting factor, with Land > 0.5 having the lowest error and performing best, followed by Sea >0.5 with the second-lowest errors followed by Ice >0.5 with the highest errors. Of note is that the Land >0.5 subset performs better than the Full case, indicating that, depending on the application, it can be beneficial to only consider soundings over mostly snow- and ice-free land.

#### 3.3 Seasonality






In the case of the tall towers network we assumed undisturbed operations during wintertime, though fluxes are lower during this time. Therefore, despite similar sampling sizes and errors, we observe on average a lower detection level twice as high as in summer (Fig. 8) which is likely related to lower wintertime fluxes (Fig. 1).

TROPOMI, being a passive sensor, has limited to no wintertime observational capabilities in the high latitudes (Fig. 7), and therefore detection limits display a strong decline during the winter (Fig. 9). As a result, detection limits increase on average by a factor of 4.8 during winter. We observe the lower detection limits of the *Sea* and *Land* subsets increasing faster from summer to winter than those of *Ice* resulting from the increasing sea ice and snow extent (not shown), and thus a relatively larger number of samples under these conditions.

MERLIN's active sensor can measure in the absence of sunlight; however, during the winter the majority of the domain is covered by snow and ice, which has a low reflectance in the shortwave infrared and substantially increases the random error. Therefore we still observe a 2.4-fold seasonal increase in the lower detection limit (Fig. 10). While this relative increase is smaller than in TROPOMI's case, the absolute lower detection limits are higher. The Land >0.5 case has been shown to have the lowest detection limits, but in wintertime the *Full*, Ice > 0.5 and Sea > 0.5 cases outperform it. This indicates that during spring, summer and early autumn the Land > 0.5 subset functions best while for the rest of the year the *Full* subset yields better results. In general, masking high error regions can improve overall performance on the metric considered here.

Figure 8: Contour plot of the Yedoma CH4 flux detection limit of the tall tower network. Shown for the 28-day bin sizes which retains most of the temporal variation. The top panel shows results for the pure detection limits (TTi) scenario, while in the bottom panel detection limits are given including transport modelling errors (TTf). Non linear y-axis shows flux enhancement (Fe), on the X-axis

the date (centre of 28-day bins). Colours and isolines indicate the number of tall towers that detect a significant difference ( $q \le 0.05$ ) between the natural and enhancement scenarios. Note that the peak of the emissions was during September.

Figure 9: Contour plot of the Yedoma CH4 flux detection limit of TROPOMI, showing the Full case. Shown for the 28-day bin sizes which retain most of the temporal variation. Non linear y-axis shows flux enhancement (Fe), on the X-axis the date (centre of 28-day bins). Colours and isolines indicate the detection limits as statistical significance (q-value) of the differences between the baseline and enhancement scenarios. Note that the peak of the emissions was during September.




455

Figure 10: Contour plot of the Yedoma CH<sub>4</sub> flux detection limit of MERLIN. Shown for the 28-day bin sizes which retain most of the temporal variation. The top panel shows pure detection limits under the union of the Land >0.5 and Full subsets, the bottom panel the detection limits of Full including high transport modelling errors. Non linear y-axis shows flux enhancement (Fe), on the X-axis the date (centre of 28-day bins). Colours and isolines indicate indicate the detection limits as statistical significance (q-value) of the differences between the baseline and enhancement scenarios. Note that the peak emissions were during September.

#### 4 Discussion

# 4.1 Methodological aspects

In the scenario presented here,  $CH_4$  fluxes from Yedoma were uniformly increased by a single, homogeneous Fe factor across the domain and time. However, this is unrealistic in the sense that rapid localised thawing of Yedoma may result in localised increased  $CH_4$  release that may be heterogeneous in space and time. While higher flux magnitudes would have a lower detection limit, more localised fluxes or temporally asynchronous fluxes would require higher detection limits. It is therefore reasonable to assume that these opposing factors might balance out over time and space. How this would affect detection limits could be quantified only in additional OSSE runs, but the definition of such detailed scenarios were beyond the scope of this investigation.

- There are limitations to the degree by which random errors, either based on previous studies or expert knowledge, are applicable and transferable. The transport modelling error characterization of tall towers is based on a study focused on the European tall tower network (Bergamaschi et al., 2022), which consist of a far denser network of tall towers than is present in the Arctic, which in turn may indicate an underestimation of the error in our study. However, methane fluxes (especially from anthropogenic sources) are higher in Europe than in most of the Arctic, and spatially heterogeneous, which would indicate an overestimation of the error in our study. To which degree these compensate each other is uncertain.
  - The TROPOMI random errors and cloud-screening were based on a best-fit to actual retrievals from Schneising et al. (2019). Despite this, the number of "good" retrievals for a given year is double that produced by our sampling due to the limited spatial and temporal resolution of our model. However, after spatio-temporal binning to account for correlation between measurements, these differences are largely mitigated.
- In this experiment we applied the random errors to the synthetic signals of both the nature run and the perturbed run. This setup is based on the premise that a baseline is built on past monitoring, which therefore implies similar uncertainty in our prior knowledge of the system. With a large enough dataset, such as e.g. a baseline set over multiple years, random errors should, by definition, average out to zero. Thus, an argument could be made that the baseline runs should not have these error terms. However, when we also consider interannual variability in both transport and fluxes, we are of the opinion that including the error terms in the baseline is more realistic.
  - To set detection limits, we performed an array of t-tests corrected by a test for false detection rates (FDR). The combination of t-test with a p threshold at 0.05 and an FDR correction with a q threshold at 0.05 is a fairly strict measure, especially since the FDR correction decreases the statistical power slightly. More lenient cutoffs would result in slightly better detection limits, although at an increased uncertainty. There are methods (Lai, 2017) to better fine tune the FDR cutoff which can be considered in future work.
  - We also note that this analysis does not fully take into account some of the benefits of the satellites: TROPOMI and MERLIN operate at a higher spatial resolution than our model runs, and while that may not directly aid in monitoring large scale processes, it is certainly a benefit that should not be overlooked, especially if the methane emissions were to happen at very localized scales that are smaller than our model resolution  $(0.5^{\circ} \times 0.5^{\circ})$ .
- Furthermore, MERLIN's expected low systematic errors are of great importance when quantifying fluxes (Bousquet et al., 2018). Unlike random errors, systematic errors do not decrease when averaging over time and space, and result in biased flux estimates. Due to the approach used in this study, this potential strength was not taken into account in our analysis.

# 4.2 Data interpretation

To put results of this experiment in perspective we look at three future example scenarios for CH<sub>4</sub> release in the Arctic based on different Representative Concentration Pathways (RCPs) (Moss et al., 2010; Schuur et al., 2022): Low, based on RCP2.6-4.5, which assumes slow warming and slow ecosystem response; Medium, based on RCP4.5-8.5, envisioning moderate to high global and Arctic warming with moderate ecosystem and landscape response; and High, RCP8.5, high global and Arctic warming with fast ecosystem and landscape response. For each of these scenarios, CH<sub>4</sub> fluxes are expected to increase significantly over time, and are considered for current conditions, halfway through the century (2049), and end of the century (2099). Considering current boreal and arctic fluxes to be on average 40 Tg C-CH<sub>4</sub> year<sup>-1</sup> (Kuhn et al., 2021; Zhang et al.,

2016), without considering transport modelling errors on average the TROPOMI and tall tower networks are able to detect a doubling of fluxes. Therefore they will only be able to detect these increased fluxes in the Medium scenario for the 2099 emissions and the High scenario from 2049 onwards. MERLIN would detect these changes in the High scenario from 2099. If we aim to allocate these flux increases to their respective sources by inverse modeling, then MERLIN's detection limits will allow this in the High 2099 scenario. Considering similar detection limits and largely similar challenges in transport modelling, TROPOMI would follow a similar pattern. The tall tower network would likely not be able to directly allocate these fluxes as a result of their high transport modelling errors. Therefore, given the expected flux increases, these systems will likely not be able to detect, let alone attribute, current changes in methane emissions from Yedoma areas. And even in the High and Medium scenarios, for which such changes could be detected, this would still be a matter of decades. This result is in line with Wittig et al. (2023), who analysed the tall tower network's ability to detect a potential 'methane bomb' emission scenario from degrading Arctic permafrost, and also found long delays in detection. With different methods these studies arrive at similar conclusions, emphasizing the robustness of these results.

It is possible that multi-year monitoring of peak fluxes (e.g. summer and autumn) could expose significant differences sooner at the cost of seasonal and spatial distinction. However this would require a reliable baseline trend and would not be informative about the source of the change. Given that a flux enhancement of 1.58 leads to a detectable enhancement for the entire network of tall towers, it is clear that these signals are quickly mixed throughout the entire domain, reaching all towers. However, when including transport modelling errors, this increases to 32.9. This indicates that while the signal reaches all towers, attribution is highly dependent on tower placement. If the goal is to link changes to relevant processes, a far denser network would be required. To properly guide such tower placement, future studies should aim to include site-specific transport modelling into the analysis and network optimization.

#### 4.3 Outlook






There is still ample opportunity for improvements to these monitoring systems. For the tall tower network, the transport modelling errors are the main crux. While improvements to the transport models themselves can partially solve this, a denser monitoring network would go far in this regard, not just in the Arctic since inversions are typically performed on global scales. Monitoring and maybe more important utilising co-emitted species may also aid in improving the inversions. With satellites we can expect to see a continued improvement in sensor quality. But especially in the Arctic they lack ground validation with fairly sparse TCCON and COCCON networks. Additionally specific regional retrieval product can be created since for example filters that make sense in a high flux well sampled region may be detrimental in one with low fluxes and few samples. Further, of note is that in this analysis we do not leverage the combined strengths of these systems. The precise measurements of the tall towers can distinguish between small changes, while the two satellites have excellent spatial coverage and resolution. While TROPOMI performs better than MERLIN in summertime (while disregarding systematic errors, as in this study), MERLIN is able to take samples in partially cloudy and dark conditions, though often at a lower precision. Retrievals from cloud tops, including cloud-slicing approaches (Ramanathan et al., 2015), may also be possible, though are not considered here. Since these systems therefore partly compensate for each other's weaknesses, a multi-stream data assimilation system can produce results better than the sum of its parts (Houweling et al., 2017). An essential component of such a system would be an extensive CH<sub>4</sub> flux network, which has been shown to be lacking in the high Northern latitudes (Pallandt et al., 2021; Peltola et al., 2019). Future studies may explore the potential of a coordinated, diverse observing portfolio to monitor such sudden emissions and changes to the northern high-latitude carbon cycle.

## 5 Concluding remarks

- In this study we presented results from an OSSE system based on GEOS-5 nature runs, to perform signal detection experiments and demonstrate the value of top-down GHG monitoring systems across the northern high latitudes. Using this system, we are able to simulate and compare detection limits of tall towers, passive and active satellites. This signal detection experiment is a first step in a larger effort to quantify the capability of high-latitude top-down networks for monitoring changes, and to a degree, warning society of sudden and profound changes in the carbon cycle as a result of climate change.
- Using our OSSE framework, we specifically targeted a scenario in which Yedoma thaw causes increased CH<sub>4</sub> release from soils to the atmosphere. We find that the tall tower network is capable of detecting the smallest flux increases tested (at a factor 1.07). Though, when relating changes to local processes the tall tower network struggles, as the lower detection limits rise to a flux enhancement factor of ~32.9 for the entire network. Minimum detection limits for the tested satellites are higher than for the best of the tall tower network, with a required flux increase approximately one and a half times larger for TROPOMI and threefold in MERLIN's case. MERLIN's ability to consistently take measurements during the Arctic winter is somewhat offset by the increased error as a result of snow and ice's low reflectance in the shortwave infrared. The transport modelling error scenarios of the MERLIN run show a relatively small increase in lower detection limits. We find these three systems will

only be able to detect changes on the scale of Yedoma thaw in the higher emission scenarios, and typically only after emissions have risen significantly over time. Longer time series can alleviate this issue to some degree at the cost of reduced temporal resolution. Furthermore, we propose an expansion of the tall tower network, and advise on an increased focus on the development of multi-stream data assimilation systems, since optimally leveraging the strengths of each of these observing systems shows great promise.

#### 6 Code and data availability

Code and data is uploaded to the EDMOND database (Pallandt et al., 2024) which will be made public on acceptance, reviewers can request access already.

## 7 Author contribution

MP was involved in all aspects of this research except Funding acquisition and Supervision. MG and AC were involved in Conceptualization and Supervision. MG, AC, LO and JM to Methodology. AC, LO and JM contributed to Data curation, Software and Validation. JM contributed in Formal analysis. MG and LO performed the Funding acquisition and Project administration. MG, AC, LO and JM contributed to Writing - review & editing.

# 8 Competing interests

535

At least one of the (co-)authors is a member of the editorial board of Atmospheric Measurement Techniques.

# 8 Acknowledgements

This work was supported by the Max Planck Society, NASA Goddard Space Flight Center, through funding by the European Commission (INTAROS project, H2020-BG-09-2016, grant agreement no. 727890), also by the European Research Council (ERC) under the European Union's Horizon 2020 research and innovation programme (grant agreement No 951288, Q-Arctic), and by the European Space Agency through the AMPAC-Nett and the GHG-CCI+ projects (contract Nos. 4000137912/22/I-DT and 4000126450/19/I-NB). This work used resources of the Deutsches Klimarechenzentrum (DKRZ) granted by its Scientific Steering Committee (WLA) under project ID bd1231. Both M.P. and A.C. were partially supported by Universities Space Research Association Goddard Earth Sciences Technology and Research (GESTAR) and Global Modeling and Assimilation Office at NASA GSFC. A portion of this research was also carried out at the Jet Propulsion Laboratory, California Institute of Technology, under a contract with the National Aeronautics and Space Administration (80NM0018D0004). M.P. and A.C were also supported by the NASA Arctic-Boreal Vulnerability Experiment (NNX17AD69A) and OCO Science Team program (80NSSC21K1068). We would especially like to thank Luana S. Basso at MPI-BGC/BSI for her review of this manuscript.

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
