# Peer review of "If the Yedoma thaws, will we notice? Quantifying detection limits of top-down methane monitoring infrastructures"

_EGUsphere, 2025_

## Referee Comment (RC2)

Review on Pallandt et al., egusphere-2025-604

**General comments**

The study examines possibility of current and future observations to detect potential CH4 emission increase from Yedoma permafrost in the Arctic. It addresses important issue of whether we can detect any changes in Arctic CH4 emissions in future. Permafrost in the Arctic is in thread of thaw due to fast global warming and Arctic amplifications, from which a vast amount of carbon could be released. To examine this point, the study used GEOS Earth System Model based Observing System Simulation Experiments (OSSE) with scenarios with varying emission enhancement factors (i.e. increase in emissions from original values) and assimilating data from ground-based stations, TROPOMI and MERLIN satellites, and their synthetic data. The authors have chosen 2010 as baseline study year, which means that some ground-based stations have actual observed data, but not all. In addition, TROPOMI and MERLIN experiments are purely based on synthetic data and their generated uncertainties. The results showed that it is challenging to detect potential changes in CH4 emissions from Yedoma with current and future planned observation networks and current accuracy/precision of the GEOS model. This is worrying and raises challenges we must address urgently. I therefore recommend the paper has potential to be published, but would like to address a few points for revision.

- Considering that the scope of AMT, I would like to see more details of the specifications of the measurement systems, especially regarding differences between TROPOMI and MERLIN. It is unclear from current manuscript that why some differences arise, and how much of them are due to design of these instruments. Do they measure same quantity? Why uncertainties are difference in these two? For example, you mention that "snow and ice negatively affects the precision of both instruments, especially MERLIN" (L277) – why does this affect MERLIN more than TROPOMI? Why TROPOMI has no reliable way of sampling over open sea? Note also that TROPOMI is name of an instrument while MERLIN is name of a mission. Although it is readable in the current form, please revise and use the terms properly in the texts.

- As shown, the results of the study depends very much on generated uncertainty estimates (both of observations and transport model). Although authors try their best to guess the uncertainties, the results has to be interpreted with much care. Therefore:
  - the authors must make it clear that the study does not tell exactly the signal detection limit of the measurements/satellite retrievals, but rather of a transport model and on regional scale.
  - I would like to see more discussion on how the assumed uncertainties are comparable to original data (for ground-based stations and TROPOMI), other studies regarding the transport model (e.g. GEOS vs other models) and total uncertainties (e.g. those used in atmospheric inversion studies) – with focus on the Arctic regions.
  - Why did you test different transport model uncertainty for the MERLIN case, but not for the TROPOMI case? Do I understand it correctly that you did not add any transport model uncertainties for the TROPOMI case? If possible, I would like to see additional simulations with transport model uncertainties added to the TROPOMI case. In addition, please present results from the MERLIN case with different transport model uncertainties – I see that you compare between nature run and high uncertainty scenarios, but not between the two uncertainties… or did I miss something?
  - Please also mention what improvements are needed to reduce these uncertainties, especially regarding retrieval methods and transport models in e.g. Section 4.3.

**Specific comments**

Section 2.2: Could you add a table with a list of sites with site information (name, location), sampling method, sampling height, sampling time (if this varies per site; see my comments below), measurement error and total error (but if a uniform transport model was assigned to all sites, then total error would not need)?

L169-172: This is a bit unclear to me. You said that your database is limited to local time afternoon, but on the other hand, it is said that sampling was done during 9:00 to 19:00 LT. Could you clarify this point? Does the sampling time vary for each site?

L175-181:
- Here are three different error terms: "measurement error", "gaussian random error", "transport modelling error". However, without descriptions of them earlier, it is not easy to understand those terms and their differences. Please add descriptions.
- It is not clear what is meant by "This gaussian random error was scaled with a 95th percentile (P95) at 2 ppb ($\mu$ 0 ppb, $\sigma$ 1.02 ppb)". You said "measurement error" was 2 ppb, but then the sentence explains "**this** gaussian random error". Do you mean e.g. "In this scenario, the gaussian random error was…"?
- I am confused with terms "95th percentile ($P_{95}$) at 2 ppb ($\mu$ 0 ppb, $\sigma$ 1.02 ppb)" and "30 ppb ($\mu$ 0 ppb, $\sigma$ 44.5 ppb)". Do you mean that you assigned random errors drown from normal distribution with mean 2/30 ppb and standard deviation of 1.02/44.5 ppb? Could you use more proper notations/terms? What does "95th percentile ($P_{95}$)" do…?
- For $TT_f$ scenario, do you mean that total observational error (measurement + transport model errors) was drawn from the normal distribution with mean 30 ppb and std 44.5 ppb, or is this just the transport model error?
- For how was the standard deviations ($\sigma$) set, i.e. why did you chose the values 1.02 and 44.5?

L194-195: What assessment did you do? Do you mean that albedo as a dominant factor for explaining variability in retrieval uncertainty?

L228: "We filtered out all fully clouded soundings"
Did you use same cloud screening as the one used to generate the TROPOMI soundings?

Regarding Sections 2.3 and 2.4:
- Please also describe how you have generated CH4/XCH4 values.
- You have described how the random errors are generated, but do you also add some systematic bias on top of it, or do you assume the systematic biases to be zero?
- What did you do regarding averaging kernels and pressure weighting functions for the gridded datasets?
- Please consider unifying the error terms used, and synthesise with terms in Section 2.2. I guess with precision, you mean "measurement error". Is there specific reason why the different term is used here?
- Could you consider adding plots showing spatial distribution of generated concentrations and uncertainties?
- Did you add any transport modelling error to the TROPOMI generated dataset? If not why?

Section 2.4: You have named two error schemes with abbreviation for the surface inversions, but not for MERLIN. Why? Just to improve readability, it could be useful to add such and use it in the main text similarly to $TT_i$ / $TT_f$ notations.

L242:"Here a, b and c are constants which were set to 20, 0.2 and 70 respectively to match Fig. 2 of Bousquet et al. (2018);"
Sorry, I do not quite understand why you chose those values, and what each variable means, even after looking at Fig. 2 of Bousquet et al. (2018), which shows maps with varying values globally. Could you add short descriptions of the variables and a bit more detail why you chose those values?

Eq. 3: Where does "142" come from, and why do you need this scalar?

In Eq. 3 and L242, does "*cfrac*" mean same as "*cf*" in section 2.3? If so, please use the same term.

L257: How did you come up with the lowest Fe of 1.06? Why you chose 80 steps in between? Could the steps be smaller or larger?

L258: Do you examine only after $1^{st}$ March when you start increasing the emissions, i.e. first bin with size 7-day is $1^{st} – 7^{th}$ March?

L289-292: Is this the case without transport model uncertainty? Throughout the result section, please make it clear which scenario you are talking about – not only of the datasets, but of uncertainties.

L319-321: I do not quite understand. Are you arguing that with smaller sample sizes, the detection limit is lower in general? But then it does contradict with the findings that detection limit is smallest in the *Full* dataset. In case of ground-based stations, I kind of understand, as it is the condition that "at least one site detect changes". But in the satellite case, the dataset is spread over the region, and there are data not close to Yamada as well. I wonder how much the number of samples in general can explain the differences in the detection limits.
- Why smaller sample size necessary lead to lower detection limit?
- Number of soundings over ice is much larger than those over land in March-April.

Section 3: Earlier, you mentioned that "we focus on the 112 day bin" (L278), but it seems that sections 3.3 focus on 28-day bin. How about Section 3.2 which bin size results are you talking about?

**Comments to Figures and Tables**
Figure 2:
- Please add also scenario when lowest Fe was used.
- Either as a subpanel or another figure, please add time series of baseline Yedoma wetland emissions and total emissions in a study domain. Please also consider adding generated $XCH_4$ values.

Figures 3, 6, 7, 8:
- "Non linear y-axis of all 80 Fe steps." I think this could be slightly misleading as y-axis ticks do not show all 80 steps. Perhaps rewrite simply as e.g. "Non linear y-axis shows flux enhancement (Fe)"?

Figure 5: Does this show number of original data or those processed and used in OSSE?

Fig. 6-8: Why x-axis starts in January when emissions should be the same in natural and enhanced scenarios? This is also related to my question regarding L258.

Y-axis labels in Fig. 3 & 6-8: Could you consider modifying them as e.g. "Flux enhancement factor (Fe)"? I think it would be more informative this way for busy readers.

Figure 7 and 8 captions: "the q-value of the comparison between the baseline and enhancement scenario".
Could you consider modifying the phrase as e.g. "the statistical significance (q-value) of the differences in the detection limits between the baseline and enhancement scenarios"?

Table 1:
- This shows statistics/values not of the original data, but generated values. Please indicate it clearly in the caption.
- What is "mean random error" exactly? Does it contain also transport model error, or do you mean measurement error (i.e. "precision" in Section 2.3 and 2.4)? Please unify terms with the main texts to avoid confusion.
- Could you also consider adding mean $XCH_4$ mole fractions, and not only the errors?
- I recommend you remove the last two rows from this table, and make another separate table regarding transport model errors added to the surface, TROPOMI and MERLIN data.

**Technical comments**
L228: "Total column soundings were performed" → "Total column soundings were generated"

L367: "not unreasonable" → "reasonable"

L412: "transport errors" → "transport modelling errors"

---

## Author Comment (AC1)

We would like to thank both reviewers for taking the time to review this manuscript and their helpful and fair comments. As suggested, we reran the model based on updated TROMPOMI products, updated most figures and added a few new ones, and in general expanded the text both to clarify the method and choices made as well as put it and the nature of the OSSE in context. This greatly improved the manuscript and we hope the reviewers agree. Below we answer (in purple) the questions and comments of the reviewers (in green).

Kind regards,
Martijn

Reviewer 2

Considering that the scope of AMT, I would like to see more details of the specifications of the measurement systems, especially regarding differences between TROPOMI and MERLIN. It is unclear from current manuscript that why some differences arise, and how much of them are due to design of these instruments. Do they measure same quantity? Why uncertainties are difference in these two? For example, you mention that "snow and ice negatively affects the precision of both instruments, especially MERLIN" (L277) – why does this affect MERLIN more than TROPOMI?
We have greatly extended the introduction to detail the differences between the two instruments. These are now sections 1.2.1 TROPOMI and 1.2.2 MERLIN
Why TROPOMI has no reliable way of sampling over open sea?
TROPOMI can measure over open sea, but only when the pixel is located sufficiently close to the sun glint location, providing sufficient signal. While the instrument does not point towards the glint location (as OCO-2 does, for instance), its broad swath, with off-nadir viewer zenith angles of up to around 60°, enables coverage over much of the earth's oceans. The measurement geometry that enables glint measurements is not only a factor of viewer zenith angle, however, the solar zenith angle also plays a role (there's a nice overview of this in https://doi.org/10.5194/amt-17-863-2024). In practice, this hardly occurs at latitudes higher than about 40° N (or below 40° S), as can be seen in Figure 12 of amt-16-669-2023.pdf.

Note also that TROPOMI is name of an instrument while MERLIN is name of a mission. Although it is readable in the current form, please revise and use the terms properly in the texts.
We already specified that TROPOMI is the name of the instrument of the Sentinel-5 Precursor mission and now clarify that MERLIN is both the name of the instrument and the mission.

• As shown, the results of the study depends very much on generated uncertainty estimates (both of observations and transport model). Although authors try their best to guess the uncertainties, the results has to be interpreted with much care. Therefore:
◦ the authors must make it clear that the study does not tell exactly the signal detection limit of the measurements/satellite retrievals, but rather of a transport model and on regional scale.
The precision of the measurements considered here, either from tall towers or from satellite, is measured in mixing ratios (ppb). However, we were interesting in measuring the sensitivity of the system to a flux signal once it was transported and mixed in the atmosphere. There is no way to carry out such a study without using a transport model to convert the fluxes to a mixing ratio (or concentration) in the atmosphere. While the magnitude of this flux signal could simply be expressed as a difference in concentration, and compared to the single measurement precision of a given instrument, this does not

accurately represent how real signals are interpreted in atmospheric data. In reality, signals are assessed over space and time, at times including averaging over different temporal and spatial scales, to get a robust result from the inherently noisy atmospheric signals. Because of atmospheric transport, a flux signal from a given location can be measured in mixing ratio measurements downwind, akin to an emission plume. Thus, this study assesses the results on a regional scale, with the use of a transport model, to represent how such signals might be seen in the atmosphere.

◦ I would like to see more discussion on how the assumed uncertainties are comparable to original data (for ground-based stations and TROPOMI), other studies regarding the transport model (e.g. GEOS vs other models) and total uncertainties (e.g. those used in atmospheric inversion studies) – with focus on the Arctic regions.

The uncertainties simulated for the synthetic TROPOMI measurements are based directly on the reported uncertainties in the WFMD V2.0 data (see Supplemental Figure S3). For ground-based stations the 2 ppb measurement precision set by the WMO is used. As explained from L230 of the revised paper, "This precision error term has a gaussian distribution and is scaled in such a way that 95% of this distribution falls within this -2ppb to 2ppb range ($\mu$ 0 ppb, $\sigma$ 1.02 ppb). This error represents a theoretical detection limit of an atmospheric signal, including the ability to detect a change, but excluding an attribution of the source of the signal." This is where an estimate of the transport error, or model representation error is required, which is typically much larger than the measurement error itself for in-situ measurements.

A value of 30 ppb was used, based on a value used in a previous study focussing on Europe. This can be compared to the root mean-squared (RMS) posterior error in simulated concentrations from a methane inversion intercomparison study also focussing on Europe (Bergmaschi et al., 2018 acp-18-901-2018.pdf), where the mean posterior RMS values ranged between 33 and 70 ppb, depending on the model. When considering studies focussing on the Arctic region, the value may even be an underestimate. Baray et al. (2021 - https://doi.org/10.5194/acp-21-18101-2021) estimated model representation errors in a methane inversion over Canada using the GEOS-Chem model with the relative residual error method (Heald et al., 2004 - https://doi.org/10.1029/2004JD005185). In this approach, the observed–modelled differences  are calculated, and the standard deviation in the residual error is used to represent the model representation error. For the Canadian domain, the mean observational error for the surface stations was found to be is 65 ppb.

◦ Why did you test different transport model uncertainty for the MERLIN case, but not for the TROPOMI case? Do I understand it correctly that you did not add any transport model uncertainties for the TROPOMI case? If possible, I would like to see additional simulations with transport model uncertainties added to the TROPOMI case. In addition, please present results from the MERLIN case with different transport model uncertainties – I see that you compare between nature run and high uncertainty scenarios, but not between the two uncertainties… or did I miss something?

No, you did not miss anything. We do not have transport modeling errors for TROPOMI since we could find no error characterization in literature that could be properly adapted for this study. This is indeed a limitation. However, in the MERLIN case we found that these are relatively minor and this likely holds true for TROPOMI as well which we highlight in the discussion. There is no reason to expect that it would be of a different magnitude than what was estimated for MERLIN, as they are measuring essentially the same quantity (see e.g. Fig. S3).  In a follow-up study we are planning a full inversion which will also allow us to investigate the effects of transport modeling errors on TROPOMI. Optimal signal detection limits including the transport model errors for MERLIN are shown in the supplement figure S4.

◦ Please also mention what improvements are needed to reduce these uncertainties, especially regarding retrieval methods and transport models in e.g. Section 4.3.

We added further discussion of potential ways to reduce uncertainties to section 4.3.

Specific comments
Section 2.2: Could you add a table with a list of sites with site information (name, location), sampling method, sampling height, sampling time (if this varies per site; see my comments below), measurement error and total error (but if a uniform transport model was assigned to all sites, then total error would not need)?
A table with all sites has been added to supplement S1. Sampling times, method, error, and transport error are uniform among all sites in this experiment.

L169-172: This is a bit unclear to me. You said that your database is limited to local time afternoon, but on the other hand, it is said that sampling was done during 9:00 to 19:00 LT. Could you clarify this point? Does the sampling time vary for each site?
Indeed, this is confusing in several ways.
We now explain that samples are taking during the day instead of the afternoon, Furthermore the window is now listed as 10:00 until 19:00 (since it was between 9:00 and 19:00) and we added the following sentence: "Since GEOS outputs have a 3-hour temporal resolution, the time offset to UTC (in hours) from where a tower is located determines if the first sample is taken at 10:00, 11;00 or 12:00. "

L175-181:
• Here are three different error terms: "measurement error", "gaussian random error, "transport modelling error". However, without descriptions of them earlier, it is not easy to understand those terms and their differences. Please add descriptions.
• It is not clear what is meant by "This gaussian random error was scaled with a 95th percentile (P95) at 2 ppb ($\mu$ 0 ppb, 1.02 ppb)". You s $\sigma$ aid "measurement error" was 2 ppb, but then the sentence explains "this gaussian random error". Do you mean e.g. "In this scenario, the gaussian random error was…"?
• I am confused with terms "95th percentile (P$_{95}$) at 2 ppb ($\mu$ 0 ppb, $\sigma$ 1.02 ppb)" and "30 ppb ($\mu$ 0 ppb, $\sigma$ 44.5 ppb)". Do you mean that you assigned random errors drown from normal distribution with mean 2/30 ppb and standard deviation of 1.02/44.5 ppb? Could you use more proper notations/terms? What does "95th percentile (P$_{95}$)" do…?
• For TT$_f$ scenario, do you mean that total observational error (measurement + transport model errors) was drawn from the normal distribution with mean 30 ppb and std 44.5 ppb, or is this just the transport model error?
• For how was the standard deviations ($\sigma$) set, i.e. why did you chose the values 1.02 and 44.5?
All these questions are linked, therefore we rewrote this section with these questions in mind to provide the needed clarity.

L194-195: What assessment did you do? Do you mean that albedo as a dominant factor for explaining variability in retrieval uncertainty?
We restructured the text to make clear this refers to Equation 1 and "Random errors in ppb, the precision, were modelled by fitting a curve to the reported uncertainties from the WFMD soundings (Figure S1), showing the strongest relation to SZA and retrieved albedo at 2.3 $\mu$m (Eq. 1)." As Eq. 1 shows, albedo and solar zenith angle are important factors in the precision.

L228: "We filtered out all fully clouded soundings" Did you use same cloud screening as the one used to generate the TROPOMI soundings?
The same cloud product was used for the filtering of both the MERLIN and TROPOMI synthetic measurements (see L242-243). When the cloud fraction (cf) was 1 for the MERLIN measurement, it was filtered. When it was less than 1, the precision was calculated as a function of cf, as given in Equation 3.

Regarding Sections 2.3 and 2.4:
• Please also describe how you have generated CH4/XCH4 values.
We now describe
For towers CH4 measurements were sampled from the bottom layer of GEOS except for the ZOTTO tower which reached into the second layer. Below we answer the question about $XCH_4$ and averaging kernels.
• You have described how the random errors are generated, but do you also add some systematic bias on top of it, or do you assume the systematic biases to be zero?
Systematic biases were originally created for MERLIN but not used since they are unaffected by the perturbation and thus have no influence on the signal detection tests. Therefore, we forewent creating them for TROPOMI. We clarified this now in the text at the end of section 2.5.

• What did you do regarding averaging kernels and pressure weighting functions for the gridded datasets?
We now describe in the text that pressure-weighted column averaging was applied to generate these modelled samples. Refer to supplement S3 for more details, including new figures S2 and S3.

• Please consider unifying the error terms used, and synthesise with terms in Section 2.2. I guess with precision, you mean "measurement error". Is there specific reason why the different term is used here?
Differences in disciplines, as far as I can say, but we now use precision throughout.
• Could you consider adding plots showing spatial distribution of generated concentrations and uncertainties?
We added new figures 3 and 4 showing $XCH_4$ and precision for TROPOMI and MERLIN for the months March and September.

• Did you add any transport modelling error to the TROPOMI generated dataset? If not why?
No, unfortunately we could not find such an error characterization in literature in a way that we could meaningfully implement it in this study. There is no reason to expect that it would be of a different magnitude than what was estimated for MERLIN, as they are measuring essentially the same quantity (see e.g. Fig. S3).

Section 2.4: You have named two error schemes with abbreviation for the surface inversions, but not for MERLIN. Why? Just to improve readability, it could be useful to add such and use it in the main text similarly to $TT_i$ / $TT_f$ notations.
We use the exact names of the 9 subsets listed in Table 1 in cursive to discuss them, we opt to not further abbreviate them. And since we have two different transport modelling errors for MERLIN they would not directly fit the tall tower notations.

L242:"Here a, b and c are constants which were set to 20, 0.2 and 70 respectively to match Fig. 2 of Bousquet et al. (2018);" Sorry, I do not quite understand why you chose those values, and what each variable means, even after looking at Fig. 2 of Bousquet et al. (2018), which shows maps with varying values globally.
Could you add short descriptions of the variables and a bit more detail why you chose those values?
It should be referring to Figure S2b in the Supporting Information document of Bousquet et al. (2018). (We updated this in the text). The numbers were part of the instrument prediction performance model from the industry prime developing the instrument (Airbus DS). All three constants are used to calculate the radiometric resolution of the instrument (the inverse of the signal-to-noise ratio), as shown in Eq. S2d and S2e of Bousquet et al. (2018). For this study, we solved for values of a, b, and c that allowed us to reproduce Figure S2b of Bousquet et al. (2018), using the reference values given therein.

Eq. 3: Where does "142" come from, and why do you need this scalar?

142 is the number of shot-pairs that are averaged over the 50-km sampling distance. By multiplying it by (1-cf), the number is reduced by the fraction that would be screened by clouds. We updated the text to explain this.

In Eq. 3 and L242, does "*cfrac*" mean same as "*cf*" in section 2.3? If so, please use the same term.
Correct, we use *cf* now.

L257: How did you come up with the lowest Fe of 1.06? Why you chose 80 steps in between? Could the steps be smaller or larger?
In this way we have large steps between the larger values and small steps between the smaller values. 1.06 is lightly lower than the lowest increase reported in Schuur et al., 2022 while, the 111 values matches a theoretical maximum (Schneider von Deimling et al., 2015; Strauss et al., 2017). This way we capture the entire realistically possible range. The exact step matters relatively little as long as the there are enough to make a meaningful distinction and the full range of interest is covered. It is not as in some ecological modelling or model optimization where an erroneous step size can cause an overshoot of the intended target.

L258: Do you examine only after $1_{st}$ March when you start increasing the emissions, i.e. first bin with size 7-day is $1_{st} - 7_{th}$ March?
We analyze the entire year but reported values such as in the text and figures 3 and 4 are optimal detection limits which correspond with the peak in concentrations as shown in figure 2 (and thus when the enhancement is in full effect).

L289-292: Is this the case without transport model uncertainty? Throughout the result section, please make it clear which scenario you are talking about – not only of the datasets, but of uncertainties.
I'm not sure what the question is here since we specifically mention that in the text:
"The impact of transport errors appears to be relatively small, with the *Low* scenario in some cases having similar detection limits as the *Full* scenario without transport modelling errors and the *High* scenario only adding an average 0.76 Fe to the detection limit." *Low, Full* and *High* reference the scenarios and their errors as described in table1.

L319-321: I do not quite understand. Are you arguing that with smaller sample sizes, the detection limit is lower in general? But then it does contradict with the findings that detection limit is smallest in the *Full* dataset. In case of ground-based stations, I kind of understand, as it is the condition that "at least one site detect changes". But in the satellite case, the dataset is spread over the region, and there are data not close to Yamada as well. I wonder how much the number of samples in general can explain the differences in the detection limits.
• Why smaller sample size necessary lead to lower detection limit?
In a t-test, sample size and statistical power are directly related. A larger sample size leads to a higher statistical power. Of course, this is only the case if all other factors, specifically the mean error remain are the same. Adding more samples does not directly improve our ability to discriminate between cases if the errors are large or the signal is small.
• Number of soundings over ice is much larger than those over land in March-April.
Yes, overall, there are more samples over ice than over land however, the errors over ice are also considerably higher.
"I wonder how much the number of samples in general can explain the differences in the detection limits."
We see that land has the lowest error but also the lowest number of samples whereas land >50 has a slightly higher error and considerably more samples (table1) and performs better (figure 4). Where the exact optimum is between the two is hard to say.

Section 3: Earlier, you mentioned that "we focus on the 112 day bin" (L278), but it seems that sections 3.3 focus on 28-day bin. How about Section 3.2 which bin size results are you talking about?
Yes, this is confusing. We updated the text to explain that unless noted otherwise, the text reflects the 112-day values. The contour plots feature the 28-day bins to preserve more of the seasonal variation.

Comments to Figures and Tables
Figure 2:
• Please add also scenario when lowest Fe was used.
We updated the figure to include those.
• Either as a subpanel or another figure, please add time series of baseline Yedoma wetland emissions and total emissions in a study domain. Please also consider adding generated $XCH_4$ values.
The emissions themselves where not sampled from GEOS and we cannot include them in a reasonable way. We added $XCH_4$ values to table 1

Figures 3, 6, 7, 8:
• "Non linear y-axis of all 80 Fe steps." I think this could be slightly misleading as y-axis ticks do not show all 80 steps. Perhaps rewrite simply as e.g. "Non linear y-axis shows flux enhancement (Fe)"?
Figure 5: Does this show number of original data or those processed and used in OSSE?
We now specified that all of these are synthetic.

Fig. 6-8: Why x-axis starts in January when emissions should be the same in natural and enhanced scenarios? This is also related to my question regarding L258.
We opted to show the entire year since this would make it easier to follow the seasonality. Note that this does not affect our ability to establish the detection limits.

Y-axis labels in Fig. 3 & 6-8: Could you consider modifying them as e.g. "Flux enhancement factor (Fe)"? I think it would be more informative this way for busy readers.
Good suggestion, we updated the figures to include this text.

Figure 7 and 8 captions: "the q-value of the comparison between the baseline and enhancement scenario". Could you consider modifying the phrase as e.g. "the statistical significance (q-value) of the differences in the detection limits between the baseline and enhancement scenarios"?
We updated the figure captions to this style, though it is the significance of the difference, not the difference in detection limits. Thus, now it reads as "the detection limits as statistical significance (q-value) of the differences between the baseline and enhancement scenarios"

Table 1:
• This shows statistics/values not of the original data, but generated values. Please indicate it clearly in the caption.
It is now written as: "Synthetic satellite retrieval subsets"
• What is "mean random error" exactly? Does it contain also transport model error, or do you mean measurement error (i.e. "precision" in Section 2.3 and 2.4)? Please unify terms with the main texts to avoid confusion.
We unified the text, and specified where it is precision and where transport errors are included as well.
• Could you also consider adding mean $XCH_4$ mole fractions, and not only the errors?
Yes, we've added extra columns with these values
• I recommend you remove the last two rows from this table, and make another separate table regarding transport model errors added to the surface, TROPOMI and MERLIN data.

Since we do not have these errors for TROPOMI, we opted do add a divider in the table for clarity instead.

Technical comments

All accepted

L228: "Total column soundings were performed" → "Total column soundings were generated"

L367: "not unreasonable" → "reasonable"

L412: "transport errors" → "transport modelling errors"

---

## Author Comment (AC2)

We would like to thank both reviewers for taking the time to review this manuscript and their helpful and fair comments. As suggested, we reran the model based on updated TROMPOMI products, updated most figures and added a few new ones, and in general expanded the text both to clarify the method and choices made as well as put it and the nature of the OSSE in context. This greatly improved the manuscript and we hope the reviewers agree. Below we answer (in purple) the questions and comments of the reviewers (in green).

Kind regards,
Martijn

Reviewer 1

General comments

The paper uses outdated versions of the EDGAR emission database and TROPOMI retrieval products (see specific comments), without a proper justification. At the very least, such a justification should be added to the paper, but the paper would be more relevant if the OSSEs were based on recent datasets. This is fair, in the specific comments we justify the use of an older EDGAR version and we redid the entire TROPOMI analysis using V2.0 of the WFMD retrieval.

Specific comments

*) page 3, line 125: it is mentioned that the model as a 0.5 degree horizontal resolution. But the horizontal extent is not mentioned, is it global or regional? Figure 1 and 5 only show latitude >~ 50N.
The model ran globally, for the analysis we only considered north of 50° North. We updated the text to reflect this.

*) page 3, line 128: Why use EDGAR v4.3.2 (which is apparently from 2017, even though the cited publication is from 2019. See https://edgar.jrc.ec.europa.eu/archived_datasets) if more recent versions of the database are available (such as the 2024 version EDGAR_2024_GHG)?

We thank the reviewer for this comment. In our case, the GEOS runs were carried out back in 2020, when EDGAR v4.3.2 GHG was the latest available version. These runs used the GEOS model configuration that is reported in Sweeney et al. 2022, ACP (although that paper itself was submitted back in 2020!). In the Sweeney et al. paper, the GEOS model simulations were extensively validated against aircraft observations over the northern high-latitudes. Since such a robust evaluation was already conducted with a specific GEOS model configuration, we decided that it would be prudent to stick to that setup for our study. We agree with the reviewer that since then more recent versions of EDGAR data are available but if we were to switch to a new EDGAR data version, we would have to go back and redo all the comparisons against independent data, make sure the model configuration and setup is robust – all of which could further delay getting this important study out to the community. Also, the GEOS model runs themselves take significant time and computing resources to run, hence we decided to stick to the previous EDGAR version rather than update it to the latest version. We would like to point out though that we do not expect our overall findings and conclusions to change since the EDGAR data provides information about agriculture and waste, fossil fuels and biofuel sectors, which are important for the total $CH_4$ signal, but not necessarily for the signal from the Yedoma region.

*) page 3, line 136: Why pick 2010 as baseline year? Later TROPOMI data from 2019 is used for the synthetic data. So you could have used actual TROPOMI data with measured cloud fractions etc..

When this was study was initiated the actual TROPOMI data were not yet available. The year 2010 was chosen to align with other GEOS runs. And since all generated data was to be synthetic, the exact year was of minor importance.

What would change if you modelled a longer time period?
In case of a burst of only one season we saw in one test run that ran for an additional year that atmospheric mixing will greatly reduce our ability in the following year without additional fluxes to detect differences. With continued enhancement we expect the lower detection thresholds to improve, though it might not necessarily improve our ability to pinpoint sources.
*) page 3, line 137: do you only amplify the Yedoma gridcells and keep the emission from the non-Yedoma grid cells constant?
Only wetland fluxes in Yedoma flagged grids are enhanced. We now clarified in the text that other fluxes are unaltered.

*) page 5, line 187: although the version numbering of TROPOMI SRON and operational retrievals could be improved, I think that v0017 is an older version of the SRON retrieval product. Lorente et al. (2022, https://doi.org/10.5194/amt-2022-255) describe important updates to the algorithm improving the retrievals. These updates have since then been incorporated into the operational (reprocessed) product. The question is therefore, why use an older version of the product?
*) page 5, line 189: Why use version 1.5 of the WFMD product? Version 1.8 has been available for years (e.g. https://doi.org/10.5194/amt-16-669-2023) while the most recent version is 2.0: https://www.iup.uni-bremen.de/carbon_ghg/products/tropomi_wfmd/index.php
Frankly, the original analysis happened so long ago that the mentioned products were up to data at the time. Though as suggested we revaluated the products and opted to redo the entire TROMPOMI analysis with the V2.0 of the WFMD product.

*) page 5, line 189/190: The sentence "Both of these products only contain successful retrievals." is incorrect. These products do contain e.g. non converged retrievals, but you can (and probably should) select only the successful retrievals with the provided quality flags.
Indeed, this was an error. We updated the text with the actual flags that were used.

*) page 5, line 199: "... fitting a curve to reported uncertainties..." Please provide a plot with this fit.
This plot is now added to the supplement (Fig. S1) and referenced in the text

*) page 5~7, sections 2.3 and 2.4: How is the model sampled for satellite observations? Is the averaging kernel taken into account?
We now describe in the text that Pressure-weighted column averaging was applied to generate these modelled samples, which is comparable to the actual TROPOMI averaging kernels and a nominal MERLIN weighting function. Refer to the supplement for more details, including added Fig. S3.

*) page 7, line 256-257: wrt. the t-test, is the test statistic used in a one or two-tailed significance test, and what about the null and alternative hypotheses?
Indeed, a few more details are required:
We compare the nature run with the Yedoma thaw scenario for each of the seven sampling and error characterizations listed above using an array of two-tailed t-tests to detect any difference with as alternative hypothesis that no detectable differences are present. We opted for two-tailed t-tests since in reality we would not know if at a certain point or time a flux would increase or decrease.
*) page 11~13, figures 6~8: why use 28-day bin sizes while in lines 277-279 (page 7) it is mentioned that "for the remaining evaluation we focus on the 112 day bin..."?

Indeed, this is somewhat confusing. These 28-day figures have a good balance between temporal resolution while still showing fairly optimal detection limits. We updated the text to indicate we do not exclusively asses the 112-day bins.

Technical corrections
Accepted all

*) page 1, line 3: affiliation of last author (Gockede) is missing.
*) page 1, line 15: please change satellites to satellite instruments.
*) page 2, line 85: since this is the first mention after the abstract of MERLIN, please provide the full instrument name in addition to the acronym.
*) page 6, table 1: fix the vertical alignent of the cell "Ice, TROPOMI" (containing the number 141461)

---

## Author Response (AR2)

We would like to thank the reviewer for their detailed review and correct all mistakes they found.

just a few technical corrections:

- \*) line 186, caption of Figure 2: replace respectivly with respectively
- \*) line 227: the text mentions "the SRON v0017 product", but this is the number from the previous version of the manuscript. The new version number in line 223 (v2.0.4) should be used instead.
- \*) line 232 contains a reference to a figure in the supplement ("...WFMD soundings (Fig. S2)..."), but it appears to be section S2, but figure S1.
- \*) The last figure in the supplement is labeled "Figure 6", while "Figure S4" would probably be better.

We also noted that figure 7 lost its colorbar in the last iteration, and updated it with the colorbar present.